# A 3.7-kW Oscillating-Amplifying Integrated Fiber Laser Featuring a Compact Oval-Shaped Cylinder Package

**DOI:** 10.3390/mi14020264

**Published:** 2023-01-20

**Authors:** Donglin Yan, Ruoyu Liao, Chao Guo, Pengfei Zhao, Qiang Shu, Honghuan Lin, Jianjun Wang, Rumao Tao

**Affiliations:** Laser Fusion Research Center, China Academy of Engineering Physics, Mianyang 621900, China

**Keywords:** high-power fiber laser, mode instability, chirped and tilted fiber Bragg grating, stimulated Raman scattering

## Abstract

Combining the advantages of high efficiency, environmental robustness, and anti-reflection behavior, oscillating-amplifying integrated fiber lasers have become popular for use in high-power laser structures in industrial applications, wherein the size of the laser source matters. Here, an oscillating-amplifying integrated fiber laser in an oval-shaped cylinder package has been proposed and demonstrated, the footprint for which only occupies an area of 0.024 m^2^ apart from the pump diode, which is much smaller than in traditional planar fiber laser packages. Numerical simulations have been carried out, which have revealed that an oval-shaped cylinder package can effectively suppress the high-order mode in large mode area fiber setups and thereby benefit the integration of fusion points and the unpackaged elements at the same time. Over 3.7 kW of transverse mode instability (TMI)-free output power has been obtained, with a slope efficiency higher than 80%. With a custom-made chirped and tilted fiber Bragg grating (CTFBG), the Raman suppression ratio is improved to reach 38 dB at peak output power. The oval-shaped design has been verified to assist with the realization of TMI suppression and improve the integration of high-power fiber lasers. To the best of our knowledge, this fiber laser has among the smallest footprints of the various fiber sources at such high-power operating levels.

## 1. Introduction

Fiber lasers fully deserve the attention paid to them by researchers around the world, due to their diffraction-limited beam quality, volume compactness, high efficiency, and superior reliability [1,2,3,4,5,6]. All these outstanding characteristics make them ideal sources in the various fields of industrial processing, such as cutting, welding, and micromachining [1,2,3], which require the continuous scaling of laser power. With the development of pump brightness and the manufacturing technique of double-clad (DC) fiber, the output power of fiber lasers has experienced a remarkable increase [7,8,9,10], while the application of fiber lasers in industrial processing has reached an unprecedented depth and breadth [11,12]. In some application scenarios, fiber lasers are challenged by harsh environments and/or strong reflecting light. Recently, an oscillating-amplifying integrated fiber laser has been proposed [13,14,15,16], which has the advantages of high efficiency, environmental robustness, and anti-reflection [13,14]. A 5-kW oscillating-amplifying integrated fiber laser has previously been demonstrated in [16], which presents a new high-power laser structure for use in industrial applications. In general, high-power fiber lasers have been laid on plane heat sinks to dissipate the waste heat [17,18,19,20]; a 4-kW fiber laser often needs to be laid on an area of 0.383 m^2^ [21], which results in a large occupying area for the laser manufacturer, necessitating more investment and more cost [22]. The cost is one of the most important calculations in industrial applications and becomes a potential obstacle for high-power fiber laser applications. In [23,24], a cylinder package has been proposed to increase high-order mode loss and suppress the onset of transverse mode instability, which can also reduce the occupied area of the fiber lasers. However, there are unbendable fiber components, such as fiber Bragg gratings and fusion splices, which cannot be laid onto the cylinder and need additional space.

In this manuscript, an oval-shaped cylinder package has been proposed and designed, which can guarantee an effective suppression of transverse mode instability (TMI) and stimulated Raman scattering (SRS) simultaneously while achieving a small footprint. By optimizing the length of the straight section numerically, the oval-shaped cylinder heat sink can offer constant diameters and optimal pump power distribution to suppress the TMI, as well as provide straight grooves to arrange the fusion points and unbendable fiber components. With a total pumping power of 4745 W, an output power peaking at 3712 W has been achieved with a slope efficiency of 80.25%, delivering a near-diffraction-limited laser beam with a measured M^2^ of 1.378 and 1.523 in the *x*- and *y*-directions, respectively. A good suppression of the SRS has been achieved with a homemade CTFBG, improving the Raman suppression ratio from 31 dB to 38 dB, at an even higher output power.

## 2. Numerical Optimization of the Oval-Shaped Cylinder

In previous research, such a cylinder has only been designed for coiling the gain fiber [24]; there has been no area available for the arranging of other fiber components. The special feature of the current laser system is the oval-shaped cylinder heat sink, which is schematically illustrated in Figure 1, as reported in Ref. [25]. Compared with a traditional cylinder, this oval-shaped cylinder includes not only semicircles but also straight sections, where the fusion points of fibers and unpackaged fiber components can be placed in the straight sections to improve their integration and compactness. The semicircles can provide a constant bend diameter to suppress the high-order modes, while the straight sections can hold certain fiber components, such as CTFBG [26], to suppress the SRS, simultaneously.

### 2.1. Numerical Modeling

Before the experiment, we conducted numerical simulations to study the influence of the radius of the semicircles, *r*, and the length of the straights, *d*, on the TMI threshold. The simulation was conducted based on the semi-analytical model used in Ref. [23]. The fiber parameters are listed in Table 1, which are typical values seen in high-power ytterbium-doped amplifiers. In the simulation, the seed power is given as 150 W, and the central wavelength of the signal is 1080 nm. The fiber used here is a commercial DC Yb-doped fiber (YDF), with a core radius of 12.5 μm and an inner cladding radius of 200 μm, and the numerical aperture (NA) is set to 0.061. With a pump wavelength of around 915 nm and a signal wavelength of 1080 nm, the absorption and emission cross-section are obtained according to the spectral characteristics of the YDF. The length of the gain fiber is 23 m, to ensure sufficient pump absorption. Since a phase-changing material is used in the cylinder for cooling, a convection coefficient of 1000 W/(K m^2^) is used. The seed power is 150 W, to fully saturate the amplifier; a strong seed into the amplifier leads to better anti-reflection properties.

### 2.2. Simulation Results

We first set the length of the straight as 10 cm and varied the coiling radius from 5 cm to 6.5 cm, with a step size of 0.5 cm. Figure 2a shows the calculated TMI threshold under different coiling diameters with different fractions of backward pump power, as used by the authors of [27]. The test setup revealed that the TMI threshold increases with the decrease in the bend radius, which agrees well with the results reported in Ref. [23]. In addition, the larger the fraction of the backward pump power, the higher the TMI threshold. Due to the conjunction when employing 915 nm of pumping and tight cylinder coiling [28], the maximum calculated TMI threshold was above 15,000 W, with a bend radius of 5 cm and a 90% backward pump power, which means that the launching pump power in the forward direction should be less than 10% for the better management of mode instability. Figure 2b shows the calculated TMI threshold using different lengths of the straight sections, with a bend radius of 5 cm. The bend radius is chosen to guarantee a high TMI threshold. It can be seen that the TMI threshold increases with the decrease in the straight length, which is reasonable because a longer straight length introduces less loss in the high-order modes. When the fraction of backward pump power is above 80%, the TMI thresholds that are under 10 cm, 20 cm, and 30 cm are all higher than 8000 W, which means that we can adjust the straight length flexibly, according to our practical needs. In the current experiment, we made an oval-shaped cylinder with a straight length of 20 cm.

## 3. Experimental Demonstrations

### 3.1. Experimental Setup

The experimental setup is shown schematically in Figure 3. The master oscillator is composed of a linear cavity, including a pair of homemade fiber Bragg gratings (FBGs) and a segment of double-cladding ytterbium-doped fiber (YDF). The high-reflector (HR) and the output-coupler (OC) FBGs are centered at 1080 nm. The 3 dB spectral bandwidths of the HR and the OC are 3 nm and 1 nm, while the reflectivities are 99% and 10%, respectively. The gain medium is a 10-meter length of commercial YDF, with a core diameter of 20 μm and an inner cladding diameter of 400 μm, and the cladding absorption coefficient of this active fiber is 0.4 dB/m at 915 nm. Three fiber-pigtailed 915 nm laser diodes are used to pump the oscillator through a (6 + 1) × 1 signal/pump combiner, which enables a maximum pump power of about 360 W. It is noteworthy that the total pump absorption of the gain medium is only 4 dB, which is specially designed to ensure that there is residual pump power launching into the main amplifier. The residual pump power launching into the main amplifier is less than 10% to achieve better management of the mode instability, as calculated in Section 2.2.

A homemade CTFBG has been employed to filter the Raman light of the output laser from the oscillator. As the CTFBG is directly scribed in the core region, the power handling capability is above 1 kW. Besides this, as the operation wavelength of the CTFBG is limited in the Raman region to around 1135 nm, the influence on the signal is negligible. The output from the CTFBG is then directly launched into the main power amplifier with a core radius of 12.5 μm and an inner cladding radius of 200 μm, respectively. The cladding absorption coefficient of this YDF is 0.6 dB/m at 915 nm, and the length of the fiber is chosen to be 23 m. In this setup, 34 fiber-pigtailed 915 nm laser diodes are coupled into the YDF through a backward (6 + 1) × 1 signal/pump combiner. Then, the boosted laser after the signal/pump combiner transmits through a cladding power stripper (CPS), to dump the unabsorbed pumping laser and the unwanted signal laser in the clad. An anti-reflection-coated endcap has been used to deliver the laser into the free-space optics safely.

The construction of the oscillating-amplifying integrated fiber laser removes the inner CPS between the seed and the amplifier. The metal packages of the FBGs and CTFBG are also removed; both facilitate the integration of the fiber laser on the oval-shaped cylinder, as well as the active cooling of the fiber gratings. As shown in Figure 4, an oval-shaped aluminum alloy cylinder was manufactured to mount the fiber for high-order mode suppression and for heat dissipation. The fiber grooves were inscribed on the outside of the cylinder. For the sake of reliability, the fusion points of the fiber and the unpackaged fiber components were placed in the straight section. The bend radius of the semicircle was set at 5 cm, to balance the optical-to-optical efficiency and the high-order mode suppression effect, which results in the footprint of the fiber laser being only 0.024 m^2^.

### 3.2. Experimental Results

Figure 5 presents the evolution of the total output power versus the pump power. The maximum output power reached up to 3.712 kW when all the backward pump power was injected. The linear fitting of the output power was also calculated and is depicted in Figure 5 as a solid red line. The first segment, shown in Figure 5 with blue dots and a blue line, is the result when only forward pump power was injected from the oscillator. The oscillator output a seed laser of 174 W with a total forward pump power of 360 W, resulting in a linear fitted slope efficiency of 53.82%. The seed laser exhibits a central wavelength of 1080.21 nm and a full width at a half-maximum (FWHM) of 1.1 nm. When the backward pump power was coupled to the laser, the output power increased with higher efficiency. The linear fitted result reveals a slope efficiency of 80.28%. The red dots agree well with the linear-fitted red line, showing no sign of the gain saturation effect near the maximum output power value, indicating further scaling ability.

The evolution of the spectrum with different output powers is depicted in Figure 6a, where the darker line represents the higher output power. Each spectrum is shifted by 1 dB in the *y*-axis for clarity. The 3 dB bandwidth of the spectrum broadened gradually along with the increase in the output power, with a maximum bandwidth of 5.3 nm at peak output power. To suppress the stimulated Raman scattering (SRS), we spliced a homemade CTFBG between the master oscillator and the power amplifier. The CTFBG was inscribed into fiber GDF-20/400 that had been hydrogen-loaded for several days. The equivalent notch filter exhibited a blocking central wavelength at around 1135 nm; the extinction ratio at 1135 nm is measured at 20 dB in low power. The Raman spectrum of the fiber laser arose when the output power reached 2580 W. However, a distinct defect was observed in the Raman spectrum, which indicates that the SRS was effectively suppressed by the CTFBG. The resulting suppression ratio of the SRS was better than 38 dB at the output power of 3.712 kW, presented in the darkest red curve. For comparison, the spectrum of the output laser without CTFBG was also recorded, which was shown in Figure 6b. The two experimental setups were exactly the same except for the CTFBG. The suppression ratio of the SRS was around 31 dB at an output power of only 3.462 kW, which also verified the effectiveness of the CTFBG in the suppression of SRS.

The output beam quality was measured according to the M2 factor, using the 4σ method at different output power levels, and the results are shown in Figure 7a. The M2 factor experienced no obvious degradation with the increase in the output power, indicating a good suppression of the high-order modes in the laser operation. The beam quality in the *x*-direction was always better than that in the *y*-direction, indicating the asymmetry of the spot caused by the presence of the high-order mode, which is excited, inevitably, by an imperfect splice point or a twist in the fiber [29]. Figure 7b shows the profile at the beam waist at peak output power; the measured M^2^x and M^2^y are 1.378 and 1.523, respectively, representing a near-diffraction-limited beam quality.

## 4. Conclusions

In summary, we have demonstrated an oscillating-amplifying integrated fiber laser with a simple optical design, mounted on an oval-shaped cylinder structure. The footprint of the fiber laser is only 0.024 m^2^, apart from the pump diodes, which is only about 1/16 of the area of traditional planar fiber laser packages. The specially designed package provides a good balance between TMI suppressibility and the compactness of a high-power fiber laser setup. The specially designed structure provides a good balance between the suppression of TMI and the improvement of integration for high-power fiber lasers. A TMI-free output power of over 3.7 kW and over 38 dB of Raman suppression ratio verify the design’s high-power compatibility.

## Figures and Tables

**Figure 1 micromachines-14-00264-f001:**
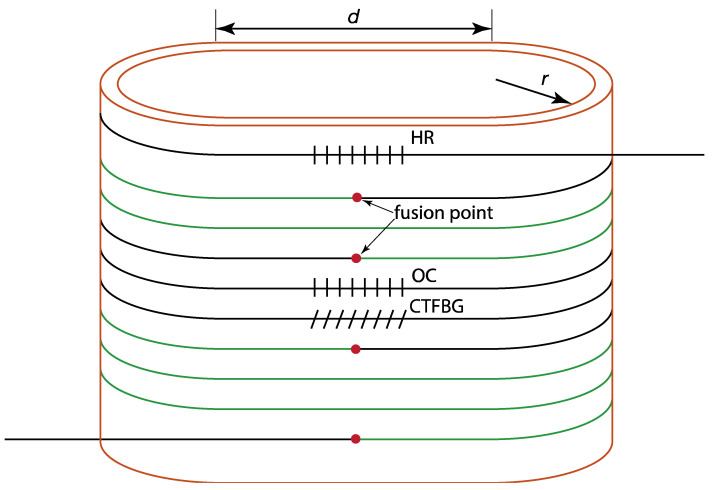
Schematic illustration of the oval-shaped cylinder. Red dots represent all the fusion points.

**Figure 2 micromachines-14-00264-f002:**
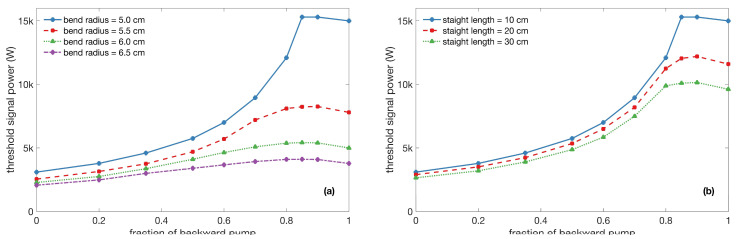
Transverse mode instability (TMI) thresholds under (**a**) different coiling radii (with a 10 cm straight length) and (**b**) different straight lengths (with a bend radius of 5 cm) with a different fraction of backward power.

**Figure 3 micromachines-14-00264-f003:**
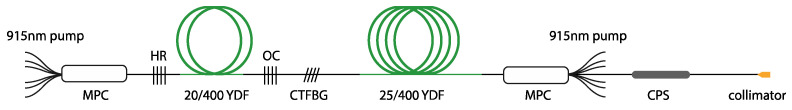
The optical construction of the fiber laser. MPC: multimode power combiner; HR: high reflector; OC: output coupler; YDF: ytterbium-doped fiber; CTFBG: chirped and tilted fiber Bragg grating; CPS: cladding power stripper.

**Figure 4 micromachines-14-00264-f004:**
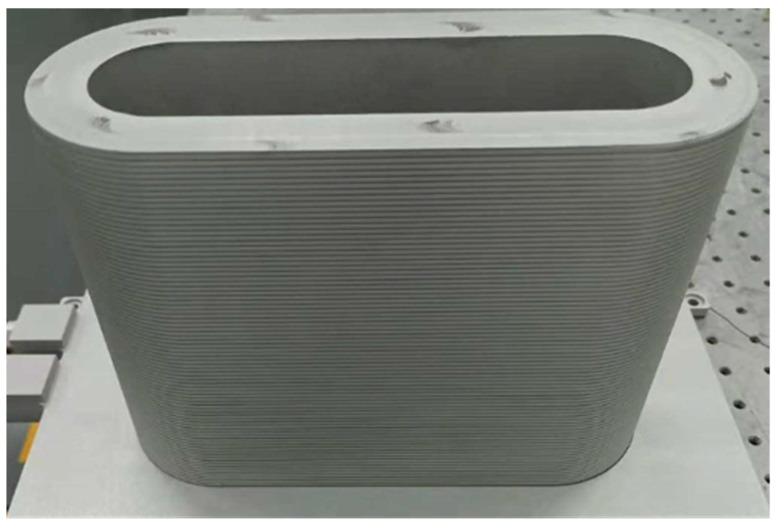
Picture of the oval-shaped aluminum cylinder acting as the heat sink.

**Figure 5 micromachines-14-00264-f005:**
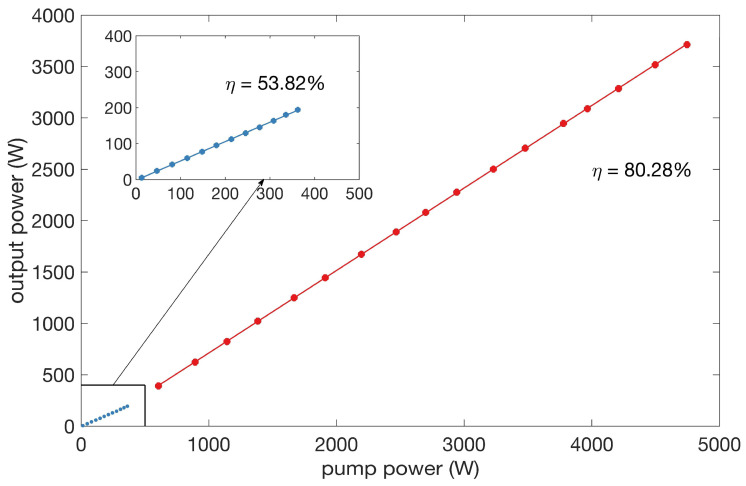
Output power versus pump power. The blue dots and blue line represent power scaling with forward pumping, and the red ones stands for those with backward pumping.

**Figure 6 micromachines-14-00264-f006:**
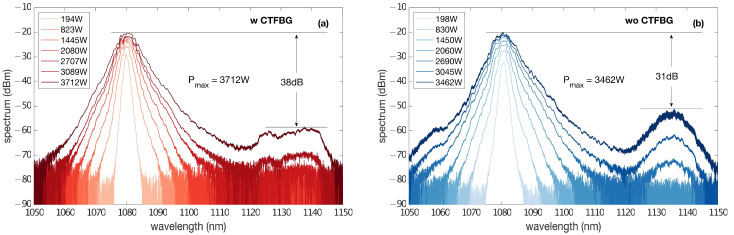
Spectra of the output laser at different powers (**a**) with or (**b**) without the CTFBG.

**Figure 7 micromachines-14-00264-f007:**
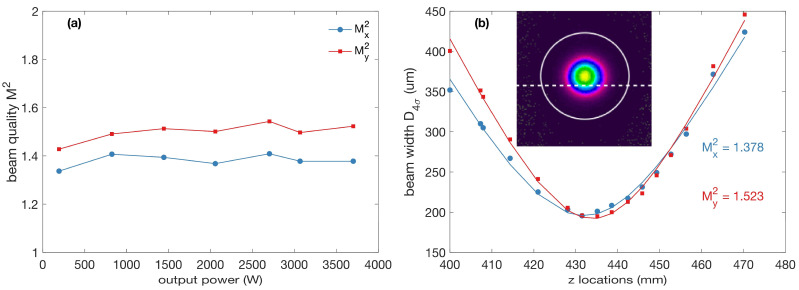
The beam quality of the laser. (**a**) The M^2^ of the output laser at different powers; (**b**) the measured M^2^ factor and the beam profile around the waist at maximum output power.

**Table 1 micromachines-14-00264-t001:** Parameters of the fiber amplifier for simulation.

Physical Symbols	Meaning	Value
$n_{\mathrm{clad}}$	Index of refraction of the optical fiber’s cladding	1.450
N.A.	Numerical aperture	0.061
$R_{\mathrm{core}}$	Radius of the optical fiber’s core	12.5 μm
$R_{\mathrm{clad}}$	Radius of the optical fiber’s cladding	200 μm
$\lambda_{p}$	Wavelength of the pumping light	915 nm
$\lambda_{s}$	Wavelength of the seed light	1080 nm
$h_{q}$	Convection coefficient for the cooling material	1000 W/(K·m^2^)
η	Thermal-optic coefficient	1.2 × 10^−5^ K^−1^
κ	Thermal conductivity	1.38 W/(K·m)
ρC	The product of density and specific heat capacity	1.54 × 10^6^ J/(K·m^3^)
$R_{N}(\Omega )$	Relative intensity noise of the input signal	−100 dBc/Hz
$\sigma_{p}^{a}$	Pump absorption cross-sections	6.04 × 10^−25^ m^2^
$\sigma_{p}^{e}$	Pump emission cross-sections	1.96 × 10^−26^ m^2^
$\sigma_{s}^{a}$	Signal absorption cross-sections	3.51 × 10^−27^ m^2^
$\sigma_{s}^{e}$	Signal emission cross-sections	4.13 × 10^−25^ m^2^
τ	Ion upper-state lifetime	0.91 ms
$P_{seed}$	Power of the seed light	150 W
$\xi_{0}$	Initial HOM content	0.01

## Data Availability

Data underlying the results presented in this paper are not publicly available at this time but may be obtained from the authors upon reasonable request.

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
