# Peer review of "A 3.7-kW Oscillating-Amplifying Integrated Fiber Laser Featuring a Compact Oval-Shaped Cylinder Package"

_micromachines, 2023, doi:10.3390/mi14020264_

Round 1
Reviewer 1 Report
This submission describes a configuration of intergated fiber MOPA packaged in a so-called "0-shpaed cylinder". CTFBG was used to suppress the possible SRS components in laser spectrum. Before my final recommendation, I would like to propose several questions for the authors to consider during their revision. My detailed comments and suggestions are as follows,
1. in Fig.6, multiple curves were demonstrated. They must denote spectra with different output power. Please add the legend in the revised manuscript.
2. It can be deduced from Fig.6 that the SRS is negligible in the oscillator and mainly caused by the amplifier. However, the CTFBGwas spliced BEFORE the amplifier. Please explain how to suppress the SRS where there is almost no SRS after the fiber oscillator.
3. About the CTFBG. Due to the special configuration of the CTFBG, there will be inevitable polarization dependant loss. Even pure randomly polarized light will become partialy polarized light after passing through the CTFBG. This change in polarization will reduce the SRS threshold in return. Please add some comments to this point in the revised manuscript.
4. I would like to know the power handling capability of the CTFBG. Please give this information in the revised manuscript.
5. In the manuscript, the seed power was so high to fully saturate the amplifier. I would like to know the efficiency with lower seed power. There must be an optimal seed power where the amplifier can still be saturated while the nonlinear effects are well controlled. Please add some comment to this point in the revised manuscript.
6. About the package. Normally, the FBG for the oscillator should be packaged with athermal design to reduce the temperature sensitivity. However, in this report, the authors destroyed the orignal package and simply wrapped the FBG pair on the cylinder. Therefore, I would like to know the wavelength stability of this laser in long-term operation with temperature rise. Please give some evidences in the revised manuscript.
Reviewer 2 Report
An experimental demonstration of a kW-class Yb-doped CW fiber laser is presented in this manuscript and one of the key features of it is mounting the whole fiber on an oval-shaped cylinder which supposedly increases threshold for higher-order transverse modes. There are also results of numerical simulation and this is the first point which raised some questions.
1. First of all, it is unusual to see just 6 points per curve because these are the points obtained from numerical simulations and some curves are not very smooth, indicating rapid changes, therefore a higher number of points is desirable. Moreover, the authors used the same model as in their previous work (cited as Ref [23]), but in the manuscript a lot of details are missing, e.g. what is the theoretical pump power (there is a fraction of backward pump, but not the overall pump power), the parameters listed in Table 1 are not known (the reader must look for a Ref [23] to find the explanations which is not acceptable). By the way, convection coefficient for the cooling liquid was 5000 W m-2 K-1 in the Ref [23], but 1000 W m-2 K-1 (5 time lower) in this manuscript, so what caused such a big change? In other words – there is no explanation of parameters and their values and how they are related to real experimental conditions. Finally, there is no clear link between results of numerical simulation and experimental conditions, because experimentally only a single bending radius and straight section were tested and for these parameters almost any fraction of backward pump would be sufficient to mitigate TMI.
2. The authors claim the used package “provides good suppression of TMI” but there is no clear proof of that. Different values of M2 parameters in orthogonal directions and both around 1.5 might be an indication that there is some fraction of higher-order modes and the authors indeed state it is “indicating the asymmetry of the spot caused by the presence of high order mode”. As a matter of fact, the M2 does increase a little bit with the increase of output power. There are other publications presenting over 5 kW of output power with very small fraction of higher-order modes and which do not use an elliptic cylinder. Ideally, the authors should unmount the fiber from the cylindrical heatsink and measure the output on a plane surface. Only then the results could be compared and it would become clear whether the use of such a package has any benefits in terms of output power and mode.
3. In the abstract the authors claim “an optical-to-optical efficiency higher than 80%”, although this is only true for the slope efficiency of the last fiber laser section. The overall efficiency is about 78% (3712 W output from 4745 W of pump).
4. A comment regarding the use of an “O-shaped cylinder”. In printed form, the letter “o” is almost round which is not the case with the real shape of the used cylinder. I would suggest to replace the term with “oval-shaped cylinder” or “elliptic cylinder”. Moreover, the authors of the manuscript presented this type of package in 2021, so they should cite their own previous work, because this is not the first time they are proposing to use it.
5. The term “occupied area” is confusing and not correct and it would be better if replaced with e.g. “small footprint”. First of all, if it is claimed as “about 1/16 area of the traditional fiber laser packages”, a citation must be presented to validate the claim and a direct comparison with some other publications using small cross section heatsinks is also required. Then, this is not the whole laser obviously as all the pump diodes and drivers are outside the cylindrical heatsink, so the claimed 0.024 m2 is just the cross section of the cavity, not the whole fiber laser.
6. “Self-developed CTFBG” shall be replace with “custom made CTFBG” or “in-house developed CTFBG”.
7. Not all references are printed using the single standard – some journals are in regular font, not italic, volume is sometimes separated using colon, but mostly using comma. In Ref [6], “6” should be used instead of “vol.6”.
8. Line 128 – two dots at the end of the sentence.
To summarize – the manuscript can be recommended for publishing after major revision and addressing the mentioned points above.
Round 2
Reviewer 1 Report
The authors have mostly addressed the questions and suggestions I proposed in last review report. Therefore, I am pleased to give my recommandation of acceptance for the revised manuscript.
Author Response
Thanks for the Reviewer's constructive suggestion.
Reviewer 2 Report
Some more notes regarding the updated manuscript.
Line 13 – the authors still did not include citations referencing footprint area of other kW-class fiber laser systems. Without citations a value of 1/16 looks just like a made-up number.
Line 83 – “cooling” should be used instead of “cool”.
Regarding the newly added sentence in Line 84 – what does it mean that a saturated amplifier allows “to achieve a better anti-reflection property”? It is not clear what authors mean by anti-reflection property? There is an anti-reflection coated end-cap at the end of the fiber, but arguably it is not related to the saturation of the amplifier.
Has horizontal axis of Figure 2 got clipped?
Regarding the added points in Figure 2 – most of the curves improved adequately, but the blue curve is still too edgy as there is something going on at very high values of fraction of backward pump. It looks like according to the theoretical model, at very small bending radiuses the calculated threshold tends to be irrationally high. The authors could check that by using e.g. 4 cm bending radius. But this is just a note regarding the model itself which probably lacks some limiting parameters.
Line 205 – should be “planar fiber laser”.
The second to last sentence is not very correct and would be better replaced with “The specially-designed package provides a good balance between the TMI suppressibility and the compactness of high-power fiber laser setup”.
Finally, suggestion for the title: “3.7 kW oscillating-amplifying-integrated fiber laser featuring a compact oval-shaped cylinder package”.
